# A Topological Selection of Folding Pathways from Native States of Knotted Proteins

**Agnese Barbensi** [1] , **Naya Yerolemou** [1,2,†] , **Oliver Vipond** [1,†] , **Barbara I. Mahler** [1] ,
**Pawel Dabrowski-Tumanski** [3] **and Dimos Goundaroulis** [4,5,6,*]

[1]    Mathematical Institute, University of Oxford, Oxford OX2 6GG, UK; barbensi@maths.ox.ac.uk (A.B.);
    Yerolemou@maths.ox.ac.uk (N.Y.); vipond@maths.ox.ac.uk (O.V.); mahler@maths.ox.ac.uk (B.I.M.)
[2]    The Alan Turing Institute, London NW1 2DB, UK
[3]    Faculty of Mathematics and Natural Sciences, School of Exact Sciences, Cardinal Stefan Wyszynski University,
    Woycickiego 1/3, 01-938 Warsaw, Poland; p.dabrowski-tumanski@uksw.edu.pl
[4]    The Center for Genome Architecture, Baylor College of Medicine, Houston, TX 77030, USA
[5]    Department of Molecular and Human Genetics, Baylor College of Medicine, Houston, TX 77030, USA
[6]    Center for Theoretical Biological Physics, Rice University, Houston, TX 77030, USA
[*]    Correspondence: Dimos.Gkountaroulis@bcm.edu
[†]    These authors contributed equally to this work.

**Abstract:** Understanding how knotted proteins fold is a challenging problem in biology. Researchers have proposed several models for their folding pathways, based on theory, simulations and experiments. The geometry of proteins with the same knot type can vary substantially and recent simulations reveal different folding behaviour for deeply and shallow knotted proteins. We analyse proteins forming open-ended trefoil knots by introducing a topologically inspired statistical metric that measures their entanglement. By looking directly at the geometry and topology of their native states, we are able to probe different folding pathways for such proteins. In particular, the folding pathway of shallow knotted carbonic anhydrases involves the creation of a double-looped structure, contrary to what has been observed for other knotted trefoil proteins. We validate this with Molecular Dynamics simulations. By leveraging the geometry and local symmetries of knotted proteins' native states, we provide the first numerical evidence of a double-loop folding mechanism in trefoil proteins.

**Keywords:** knotted proteins; protein folding; knotoids; knots; topological data analysis; bioinformatics; computational biology

## 1. Introduction

A small percentage of proteins deposited in the Protein Data Bank [1] (PDB) are known to fold into conformations that are non-trivially entangled. All the currently known knotted protein structures are catalogued in the online database KnotProt [2]. The vast majority of knotted proteins form simple knot-like structures. However, there are a few examples of proteins having more complex entanglement types [2]. Interestingly, even though they are not ubiquitous, knots in proteins have been conserved within species that are separated by millions of years of evolution [3]. This might imply that knottiness provides some advantages to proteins [4], but this theory is still being contested [5]. Besides the role of knots in proteins, there are many open questions about the pathway that the backbone of a knotted protein follows to reach its native folded state. To date, researchers have suggested several mechanisms for protein self-tying that are based on wet lab experiments, numerical simulations, mathematical theory or a combination of all the above [3,6–21].

Recently, Flapan et al. [21] proposed a novel topological model that contains as special cases two of the most popular folding theories [13,22]. This theory is agnostic in terms of protein families, and it is based on the assumption that there exist two loops of different sizes, each containing up to two twists. It is shown that there are several possible pathways emerging from the different choices of parameters of the model and it is suggested that

these pathways are enough to recover any knot-type found empirically thus far in proteins. As this model is theoretical, physical interactions and dynamics that may affect the folding pathway are not taken into consideration.

In this work, we systematically analyse the native states of all deposited positive trefoil knotted proteins, aiming to determine potential folding pathways. We focus on trefoil proteins as these make up the majority of knotted proteins. Considering that the global underlying topology of all proteins in our data set of proteins is a trefoil, the geometry of the conformation plays a key role in detecting the most probable folding motif.

Our analysis is based on KnotoEMD, a statistical distance that we define on distributions of protein backbone projections considered as knotoids. Knotoids are knot-like open-ended topological structures that extend knot theory to the case of open curves [23]. The knotoid approach is a probabilistic method that does not require an artificial closure of the protein backbone. Multiple projections of the backbone are considered to determine the predominant knotoid type [24]. It has been shown that knotoids provide a more refined overview of a protein's topology [2]. KnotoEMD is a specialisation of the earth mover's distance, i.e., the 1st Wasserstein metric [25], and it quantifies the distance between different knotoid distributions in the space of trefoil knotted proteins.

We observe the existence of a double-loop in the conformation of shallow knotted carbonic anhydrases, which suggest that their folding follows a pathway similar to the theoretical model of Flapan et al. We subsequently perform computer simulations for the carbonic anhydrase with PDB entry 4QEF. The results confirm that this protein follows a complex folding pathway involving a double loop formation at the beginning of the process, although the specific mechanisms proposed by Flapan et al. are not observed. In contrast, a much simpler geometry is revealed for other families of shallow knotted trefoil proteins and for all the deeply knotted ones, providing strong evidence against a double-loop hypothesis in these cases. This is in agreement with the current consensus that different folding pathways should be accessible by shallow knotted proteins [26,27] and with previous simulations of deeply knotted trefoil proteins [20]. We find that KnotoEMD extracts subtle geometric information from the knotoid distribution and groups together trefoil proteins by sequence similarity. This highlights the fact that KnotoEMD is a versatile and robust mathematical tool that captures subtle topological and geometric information of open-ended conformations and has potential beyond the scope of this research, as discussed in what follows.

## 2. Materials and Methods

### 2.1. The Knotoid Distribution Describes a Protein's Entanglement

Mathematically, a knot is formally defined as a closed curve in 3-dimensional (3D) space, and two knots are deemed equivalent if one can be deformed into the other without cutting and pasting the curve [28]. In order to identify the topological type of a linear knotted protein, a classical approach consists in considering its backbone as a polygonal curve and then joining the two termini with a new segment. The resulting closed curve is then analysed as a knot, and its underlying knot type can be formally determined [29]. The obtained knot type is clearly dependent on the choice of segment connecting the end-points of the curve. There are several ways to create a closed curve out of an open one in an unbiased way, in order to approximate the topology of an open curve [30]. Perhaps the most popular approaches are those where one considers several different closures at once and then takes the probability distribution of knot types as a topological descriptor of the open curve [31,32]. In such a probabilistic approach, the most likely outcome in the knot distribution defines the protein's dominant topology and characterises its global entanglement type, i.e., the topology of the entire curve. The position and the relative size of the knotted portion of the protein can be identified by the so-called knot fingerprints [33].

Recently, the introduction of new mathematical objects called knotoids inspired a more refined approach [34] aimed at detecting geometric and topological features of entangled open arcs subtler than their global knot type [2]. With this approach, the topology of

a knotted protein is described by the distribution of its planar projections' topological types [24,35], as shown in Figure 1.

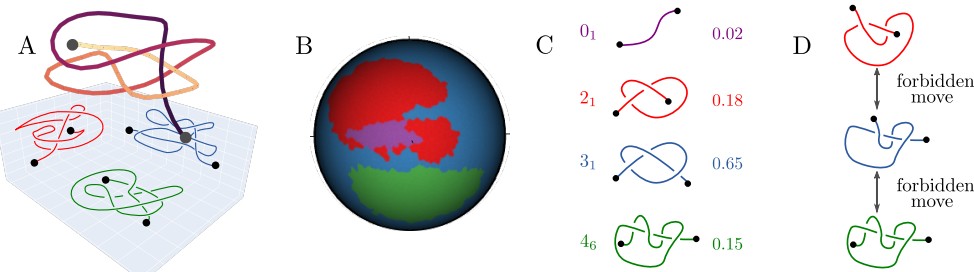

**Figure 1.** Knotoid Distribution and Forbidden Moves. (**A**) An open curve in 3D space with three different planar projections. Each projection defines a knotoid diagram, and different projections might yield non-equivalent knotoid diagrams. By considering all of the possible planar projections, we can associate a distribution of knotoids to the initial open curve. (**B**) The distribution of knotoids of the curve in panel (**A**) is visualised by colouring each point of a 2-sphere surrounding the curve according to the topological type of the corresponding projection. This picture is obtained using the software Knoto-ID. (**C**) The distribution of knotoids of the curve in panel (**A**). A minimal diagram is shown for each knotoid type. The knotoid distribution of each curve analysed in this work was computed using Knoto-ID. In practice, 5000 projections are computed for each curve, and this is known to be sufficient to properly approximate the continuous distribution. (**D**) Knotoid diagrams related by forbidden moves. Any knotoid diagram can be untangled by a finite sequence of forbidden moves.

A *knotoid diagram* is a possibly self-intersecting open curve in 2-dimensional (2D) space together with crossing information, i.e., the data on which portion of the curve is on top at each self-intersection. Two knotoid diagrams are considered equivalent if they differ by local transformations called Reidemeister moves, performed away from the endpoints [23]. A knotoid is then defined as an equivalence class of knotoid diagrams under this relation. Transformations that displace the endpoints of the diagram over or under other arcs can change the knotoid type, and any knotoid diagram can be unknotted by performing finitely many such moves. A knotoid can be interpreted as an open curve in 3D space with the additional data given by a fixed direction of projection [23,36].

Each planar projection of a curve in space yields a knotoid diagram. The shape of an open curve in 3D can thus be summarised by assigning the knotoid type of the 2D projection in each direction, as described in Figure 1A,B. This procedure assigns a *knotoid distribution* to an open curve in 3D, with the distribution given by the proportion of directions for which each knotoid is observed [24,34,35]. Crucially, the knotoid approach provides a richer representation of a protein's entanglement than any knot closure technique [2]. Note that the knotoid distribution exhibits a natural antipodal symmetry.

The existence of open ends in a knotoid diagram enables another class of moves, whereby the endpoints pass under or over portions of the curve and potentially change the knotoid type, known as *forbidden moves* [37] (see Figure 1D). Forbidden moves allow every self-intersection of a knotoid diagram to be resolved and thus to untangle the diagram, in the same way that a tangled open curve in 3D can always be untangled without being cut. Counting the minimal number of forbidden moves required to pass between knotoid types quantifies the difference in their entanglement and defines a distance, $d_f$, known as the *f-distance* [37]. Similarly, the minimal number of forbidden moves needed to untangle a knotoid measures how complex the knotoid is. This notion of complexity can be encoded in the knotoid distributions to effectively differentiate between open curves sharing the same underlying global topology, i.e., with the same dominant knot type [37].

### 2.2. KnotoEMD: A Topological Distance to Distinguish Geometric Features of Knotted Proteins

Based on the above, it is reasonable to assume that knotoid distributions and pairwise $f$-distances of intra-distribution knotoids may reveal key information about the shape of knotted proteins. Motivated by this, we introduce KnotoEMD, a distance between the knotoid distributions of open curves in 3D. This distance incorporates both the topology (i.e., the global knot type) and geometry (i.e., local features and shape) of open curves. Given two such curves, KnotoEMD is defined as the earth mover's distance [25] between their knotoid distributions, with cost matrix given by the $f$-distance. Our distance quantifies the cost of converting one knotoid distribution into another, where the cost per unit of converting one knotoid type to another is the $f$-distance between them. Intuitively, this captures the amount of untangling and tangling required to pass between knotoid distributions, and thus offers a way to meaningfully compare the entanglement of proteins with the same global topology, which was not possible with previous methods [2,24,29,31–33,38].

The KnotoEMD approach for knotted proteins incorporates the global topology of an open-ended conformation as well as its local geometric features and overall shape. For example, the presence of nugatory crossings—crossings that can be removed just by rotating a part of the curve—in a conformation will not dramatically change its global topology, as this is averaged out over a large number of projections. However, the knotoid distribution will likely differ from the same conformation but with the nugatory crossings removed. For this reason, we can consider that such geometric features of the conformation act as a confounding variable of our pipeline. Two curves are considered close by KnotoEMD if the lists of knotoids in their respective knotoid distributions are almost overlapping and with similar individual weights. For example, two curves differing only by local changes away from the endpoints, such as the ones in Figure 2C, are very close to each other according to KnotoEMD. At the same time, two sufficiently different embeddings of the same curve might be far from each other: a minimal realisation of an open trefoil, for example, would have a large KnotoEMD from any open trefoil with higher average crossing number.

### 2.3. Folding Hypotheses for Knotted Proteins

Results over the past decade arising from theory, simulations and experiments [3,6–12] reveal at least four potential folding mechanisms for trefoil proteins [20]. All of these pathways rely on the formation of a single twisted loop at one stage of the folding process, as shown in Figure 2A. They differ by the occurrence of N/C-terminus threading [13–15], loop-flipping (also known as mouse-trapping) [11,16,17] and by the presence of ribosomes and chaperones [12,18,19]. A detailed description of various folding mechanisms is beyond the scope of this paper. We point the interested reader to a review by Sułkowska and the references contained therein [20]. The single loop formation described by these folding mechanisms predicts a folded geometry resembling a somehow minimal open trefoil having only one pierced loop, as shown in Figure 2, referred to henceforth as single loop (1-L) trefoil configurations. In reality, the geometry of all proteins is far more complex than these idealised representations; however, the smoothed, coarse geometric structure of a trefoil protein should resemble the minimal, single-looped geometry proposed by the hypotheses, if the protein folds according to one of these pathways.

Flapan et al. [21] introduced a new theoretical model for the folding of knotted proteins based on loop flipping and numerical simulations of folding of complex knots [22]. A characteristic feature of this model is the formation of two initial loops, as opposed to one (see Figure 2B). Similarly to above, the formation of two loops predicted by this model should result in creating more complex, double-looped trefoil geometries. Knotting pathways in this model depend on the choice of several parameters, which regulate the number of twists in the loops and their relative positions. Twelve possible choices of parameters were identified, each yielding a different double-loop open positive trefoil (2-L) configuration. The twelve 2-L configurations are labelled by a two-letter word in $\{L, R\}$ followed by a $\pm$ and two (signed) integers, describing the relative positions of the loops and the number of twists [21]—a list of these names can be seen in Appendix B. Out

of these twelve 2-L configurations, the least topologically complex are proposed as the most likely to occur as native states of trefoil proteins, while the more complex ones are discarded [21] based on an approximation to the knot fingerprint. We observe that the twelve proposed 2-L configurations can be divided into three groups, where configurations within each group are related by local transformations, as illustrated in Figure 2C. For more details, the reader is referred to Appendix B.

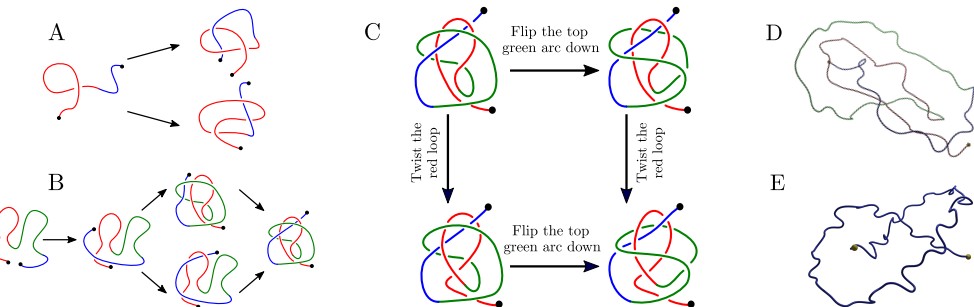

**Figure 2.** Folding Pathways for Trefoil Proteins. (**A**) A schematic representation of the single-loop folding theories. According to these theories, the formation of a twisted loop (native or not) is followed by either threading one terminus through the loop (top) or flipping the loop over the terminus (bottom). (**B**) A schematic representation of the double-loop folding theory. Two loops are formed and then approached by one terminus. This is followed by a combination of threading and flipping, in either order. (**C**) Simple 2-L configurations and moves between them. Two diagrams in the same row are related by a flipping of the green arc, while diagrams in a column are related by a twist of the red loop. Note that performing any of these moves is possible by fixing the majority of the curve and changing only one small portion. The eight other 2-L configurations are divided similarly into two groups of four. More details can be found in the Appendix B. (**D**) An example of a generated 2-L trajectory, rendered using Blender [39]. (**E**) An example of a generated 1-L trajectory, rendered using Blender [39].

### *2.4. Methods*

The folding pathways described in the previous section predict two different overarching trefoil geometries as an end result. Thus, we seek to differentiate trefoil proteins via their geometric entanglement in order to infer information about their respective folding pathways. As we are working with proteins in their native state deposited in the PDB, we work under the premise that different folding pathways leave different geometric artefacts in the knot cores, such as a single or double loop in the proteins' native conformations. With this aim, we generate suitable curves of different lengths for the 1-L and twelve 2-L configurations predicted by the two folding models, and then we apply numerous rounds of numerical perturbations to produce several hundred piecewise-linear open curves resembling the end-states of each of the pathways described by the models to compare with positive trefoil proteins. Our final data set consists of all the positive trefoil ($K+3_1$) proteins taken from KnotProt [2] and a collection of generated open curves for the 1-L and 2-L configurations (see Figure 2D,E). More details on our data set can be found in Appendices A and B. We then use the knotoid approach for each curve in the data set and compute its knotoid distribution. The knotoid distribution of each curve analysed in this work was computed using Knoto-ID [35]. In practice, 5000 projections are computed for each curve, and this is known to be sufficient to properly approximate the continuous distribution [37]. Subsequently, we compute all pairwise KnotoEMD distances between distributions in the data set. All of these computed pairwise distances are contained in distance matrices that can be found in Appendix C; for data visualisation purposes, we display our results using the dimensionality reduction technique, UMAP [40], projecting the space of proteins and/or curves with distance given by KnotoEMD to a plane, while preserving the KnotoEMD distances as much as possible.

Our results suggest that the carbonic anhydrase 4QEF is entangled in a way similar to a 2-L configuration, so we selected it as a candidate protein for simulations to further probe its folding pathway. The coarse-grained simulations were performed in Gromacs 5 [41] using a $C\alpha$ structure-based model [42] with parameters proposed by the SMOG server [43] and shadow algorithm [44]. The native structure taken from PDB was first unfolded at a high temperature to obtain the starting frames for folding. Next, 200 folding trajectories with 300–500 millions of frames each were performed close to the folding temperature, where there is an equilibrium between folded and unfolded structures.

## 3. Results

We find that KnotoEMD is capable of detecting subtle geometric information from the knotoid distribution. This is reflected by the fact that it is able to group together trefoil proteins by sequence similarity. We differentiate between double-loop and single-loop open trefoils via this geometric and topological information, and detect fine structure in the space of trefoil proteins. The local geometric features detected suggest different folding pathways for trefoil proteins. In particular, we find that the proposed double-loop mechanism is the most likely for carbonic anhydrases forming a shallow trefoil.

### 3.1. Sequence Similarity from the Geometry of Proteins' Native States

KnotoEMD is sensitive to the geometry and topology of the entire open curve, and is therefore influenced by the depth of the knot core, locations of the termini, and the structure and entanglement of its core. These features are often shared by proteins with similar sequences, and these proteins are therefore seen as similar by KnotoEMD, as shown in Figure 3B. To show this, we cluster all trefoil proteins by first computing the pairwise sequence similarity scores, using PDB, and then performing single-linkage hierarchical clustering and pruning clusters at a threshold of 30% similarity. More details on our subdivision can be found in Appendix A, and a dendrogram showing the clustering by sequence similarity is shown in Figure 3A.

The structural features of the dendrogram are reflected almost perfectly by Knoto-EMD, for example, the proteins represented by 4QEF appear to form two separate groups in the UMAP [40] projection, coherently with the fact that two light blue clusters are merged quite late in the dendrogram to form the final one. Similarly, tight clusters in the dendrogram, such as the one represented by 6QQW, are compact in the UMAP projection. Moreover, we are able to detect a range of geometries and to efficiently separate trefoil proteins by their geometric features. Indeed, by design, KnotoEMD takes into account both the dominant topological type of a protein and its overall structure; thus, it is able to detect features of open curves not captured by previously defined methods [4,32,33,45] used to compare them. In particular, it is able to differentiate between proteins with the same dominant topology and at the same time distinguish pairs of homologous proteins with almost superimposable structures differing only by a strand passage, as in the case of the pair knotted AOTCases/unknotted OTCases [4] (see Appendix C). Within this study, we investigate the geometry and topology of protein structures and compare their entanglement to knotting geometries predicted by the hypothesised knotting pathways. As there is no restriction on the length of the two ends in the knotting hypotheses we consider, the pathways really only describe the entanglement of the core. In order to make the conformations proposed by the hypotheses comparable with the entangled proteins, we clip the ends of protein backbones up to their entangled core using a top-down approach [35]. When reducing the chain to the knotted core, the sequence similarity clustering is maintained, although it is slightly noisier in this case. The complete results on pairwise distances between protein cores can be found in Appendix C.

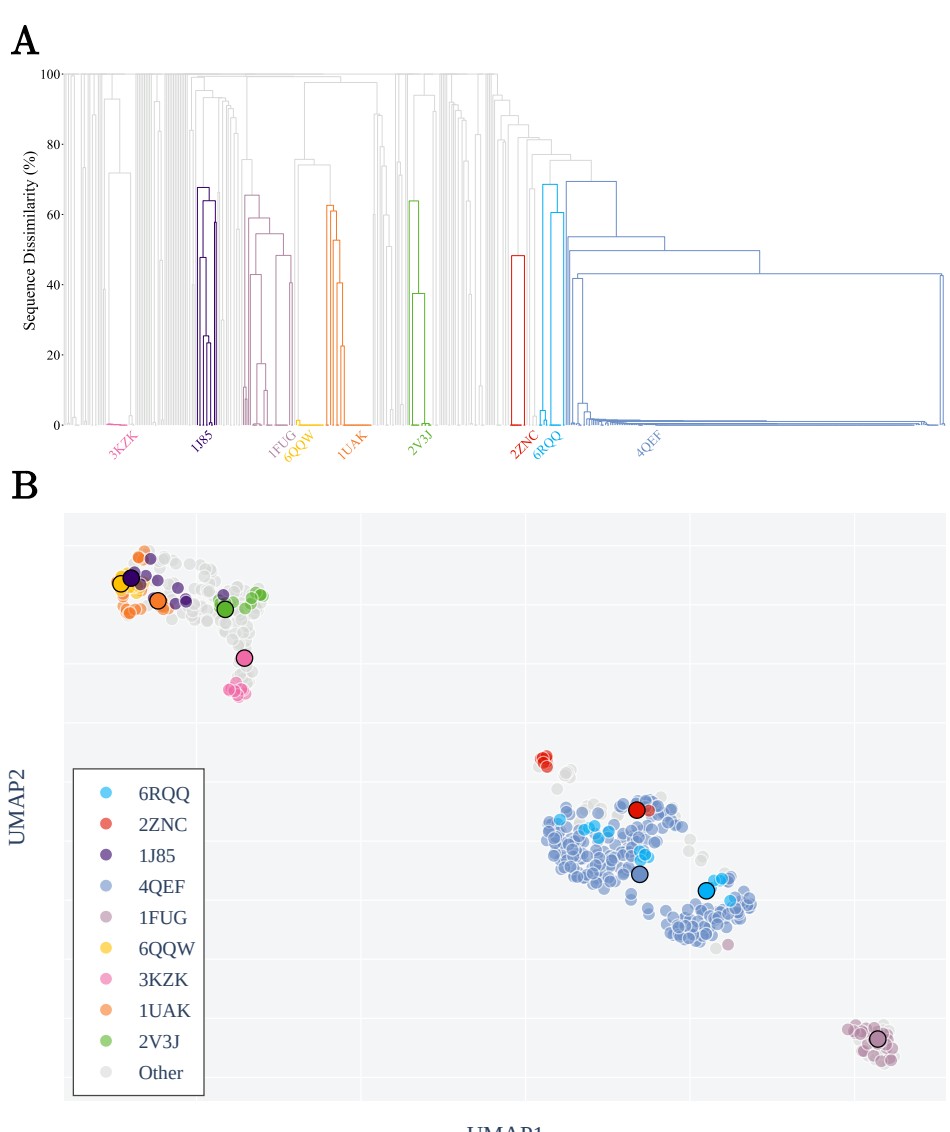

**Figure 3.** KnotoEMD clusters trefoil proteins by sequence similarity. (**A**) A dendrogram obtained from single-linkage hierarchical clustering of all trefoil proteins based on sequence similarity scores from PDB. (**B**) A UMAP projection of the space of trefoil proteins with KnotoEMD distances, when considering the entire protein backbone. Non-redundant proteins are indicated with thicker boundaries. We highlight the groups corresponding to proteins with sequences similar to 6RQQ, 2ZNC, 1J85, 4QEF, 1FUG, 6QQW, 3KZK, 1UAK, and 2V3J, as these are the nine largest in our subdivision. Note that 2ZNC and 4QEF are all carbonic anhydrases with very similar structure. Similarly, 1UAK and 6QQW are both tRNA-methyltransferases. All of the deeply knotted proteins are contained in the cluster on the left, and this cluster contains no shallow knotted trefoil proteins. Depth is a dominant factor in determining the knotoid distribution, explaining why KnotoEMD separates proteins with a very deep knot from everything else. Finer structure in the space of proteins as detected by KnotoEMD can be seen in 3D-plots available in our GitHub repository [46].

### 3.2. KnotoEMD Captures Subtle Geometric Differences between Double-Loop and Single-Loop Open Trefoils

In a recent study, Piejko et al. highlight different folding behaviours for deeply knotted and shallow proteins [26]. It is then natural to ask whether there are geometric features of trefoil proteins, other than depth, that could be used to infer further information about their knotting mechanisms.

The knotoid distribution can vary substantially for different geometric realisations of an open-ended trefoil. We quantify such differences in the knotoid distributions by computing the pairwise KnotoEMD between the cores of our 1-L and 2-L trajectories, and of trefoil proteins. As shown in Figure 4, the three groups of 2-L trajectories and the group of 1-L trajectories appear clearly separated, while 2-L trajectories within each of the three groups are close to each other. This demonstrates that KnotoEMD successfully captures subtle geometric structure, as configurations within the same group only differ by simple local moves (see Figure 2C), which only slightly impact the knotoid distribution and do not substantially change the overall geometry of the core.

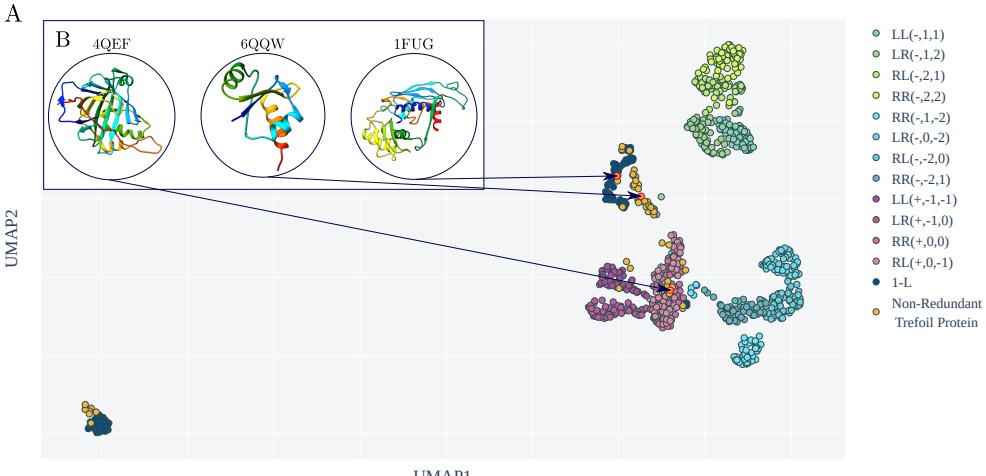

**Figure 4.** Local geometric features suggest different folding pathways for trefoil proteins. (**A**) 2D UMAP projection of the pairwise KnotoEMD between 1-L trajectories, 2-L trajectories and non-redundant trefoil proteins. Recall that 2-L configurations can be divided into three groups, in which configurations differ by small local transformations. 2-L configurations LL(-,1,1), LR(-,1,2), RL(-,2,1), RR(-,2,2) and RR(-,1,-2), LR(-,0,-2), RL(-,-2,0), and RR(-,-2,1) form the two complex groups and LL(+,-1,-1), LR(+,-1,0), RR(+,0,0), and RL(+,0,-1) the simple one. The clustering by KnotoEMD respects this subdivision almost perfectly. Similarly, 1-L configurations are clustered together, with the exception of a bottom-left group, which is separated due to oversimplification. The proteins appear distributed among the simple 2-L and the 1-L clusters. The small isolated clusters contains 1-L configurations and deeply knotted proteins whose core is oversimplified due to perturbation or noise in knot core reduction. (**B**) The knot cores of a shallow protein in the 2-L cluster (a human carbonic anhydrase, PDB entry 4QEF), a deeply knotted protein in the 1-L cluster (a tRNA-n1g37 methyltransferase, PDB entry 6QQW) and a shallow protein in the 1-L cluster (a S-adenosylmethionine synthetase, PDB entry 1FUG).

### 3.3. Local Geometric Features Suggest Different Folding Pathways for Trefoil Proteins

When analysing pairwise distances between protein and configuration cores, each trefoil protein core appears to have a knotoid distribution highly similar to either those of 1-L trajectories or of simple 2-L trajectories. This is particularly evident when looking at the distances between our generated trajectories and the list of non-redundant trefoil proteins taken from KnotProt [2], as visualised using a UMAP [40] projection in Figure 4. Our results are coherent with the fact that complex 2-L configurations exhibit knot fingerprints that are not compatible with those of trefoil proteins [21]. Indeed, trefoil proteins are distributed unevenly between the two simple groups (1-L and simple 2-L trajectories), with the vast majority having a knotted core with topological complexity indistinguishable from that of simple open-ended trefoil cores, i.e., from the 1-L trajectories. In particular, all of the non-redundant deeply knotted proteins we considered share this feature. Again, this is coherent with previously simulated folding pathways for deep trefoil proteins [6,20,26], all involving the creation of a single twisted loop. This single loop can form either in a native

state, as for the threading mechanisms, or unfolded, as in the loop flipping hypothesis. We emphasise that our computations demonstrate that the presence of two pierced loops in an open trefoil inevitably results in increased complexity of its knotoid distribution, making it substantially different from that of a simple trefoil; indeed, all three double-loop clusters appear clearly isolated in Figure 4. Therefore, our results provide strong evidence that the double loop folding mechanisms [21] can be excluded for deeply knotted trefoil proteins.

The same is true for the shallow protein 1FUG (see Figure 4), whose knot core appears very simple despite being considerably longer than the ones of deep proteins (~256 amino acids), and even longer than many shallow ones. For these proteins, the cores' low topological complexity is shown in Figure 4, where the single-loop structure is clearly visible.

Note that there are two separate clusters of 1-L trajectories, each containing some proteins. For the 1-L trajectories, this stems from the fact that perturbing might have resulted in an over-simplification of the curve. Similarly, small errors in the knot core reduction of proteins with particularly simple core might give rise to an almost trivial knotoid distribution. In both cases, this happens to curves whose actual core is very simple, so this does not affect the interpretation of the results.

There is a consensus that multiple folding pathways are possible for shallow proteins [26,27,47], and this is reflected by our results. We find that, contrary to the aforementioned 1FUG which clusters with the 1-L trajectories, all carbonic anhydrase representatives, a group of proteins with very shallow trefoil knots, are clustered with simple 2-L trajectories. Therefore, their knotoid distributions have a complexity coherent with the double-looped trefoil geometries. This suggests that a single-loop formation followed by either threading a terminus or flipping a loop over a terminus are unlikely folding mechanisms for carbonic anhydrases. A pathway requiring double loop formation, such as those described by Flapan et al. [21] is the most likely among the proposed ones, as KnotoEMD shows that the knot core is tangled in a way most similar to a 2-L configuration.

To test this hypothesis, we performed coarse-grained simulations of folding for the carbonic anhydrase with PDB code 4QEF. This protein features a very interesting 'spine' of $\beta$-sheets, which joins together different parts of the protein. In particular, one can identify a small fragment of the chain extruding from the $\beta$-sheet core, which can be interpreted as one loop, shown in green in Figure 5, as well as a long fragment surrounding the entire $\beta$-core, which could be interpreted as the second loop, shown in red in Figure 5. The C-terminal tail protrudes from the inner loop and pierces the outer loop, eventually forming the knot. The folding of this protein starts from the formation of the $\beta$-sheet core and hence, the two loops are formed at the initial stage of the folding process. The $\beta$-sheet can be formed starting either from the formation of the outer red loop, or the internal green loop, as shown in Figure 5. Once both loops are formed, the tails are then placed and the simulations suggest the existence of two different pathways, depending on which tail is placed first. Similar behaviour was observed in the case of the $5_2$-knotted UCH-L [48], which was used as a example of a protein following a double-loop folding mechanism by Flapan et al. [21]. This mechanism, although not evidently confirming the mixed threading/mouse-trap movement event, is more complex than a simple threading mechanism observed for deeply knotted proteins.

The difficulties in unequivocal confirmation of the double-loop folding mechanism may be due to the fact that the mechanism detected is the result of some evolutionary optimisation. From a thermodynamic perspective, the folding speed is limited by the free energy barrier stemming from the knot formation. In particular, threading a free, dangling tail through an already formed loop is associated with a significant decrease of entropy, which is overcome by the concomitant formation of native contacts. This entropy–enthalpy interplay slows down the folding process, and would occur twice in a 'pure' 2-L folding mechanism (i.e., exactly like the one in Figure 2B), as there are two loops which have to be threaded. Therefore, from a kinetic point of view, such a mechanism could be optimised to reduce the threading burden. One obvious optimisation would be to make the folding similar to a single-threading mechanism, as noted also in [21]. In such a case, the observed

double loop mechanism would be less evident, leaving only traces in the folding process or in the native structure. Our results indicate that these traces can be detected by KnotoEMD, giving material for follow-up studies, e.g., with mutation-based analysis.

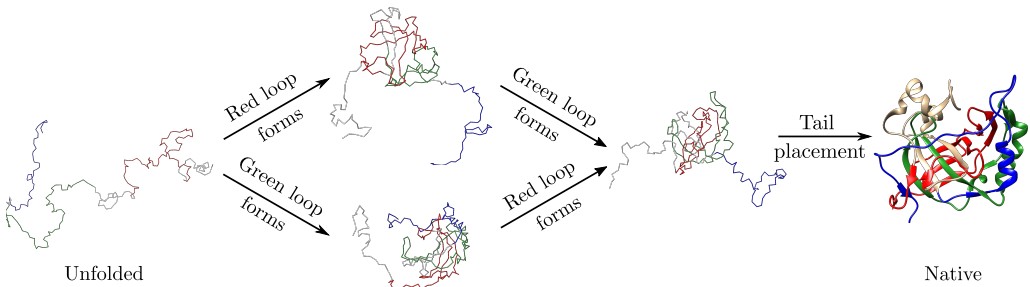

**Figure 5.** The scheme of the folding pathway of a carbonic anhydrase (PDB code 4QEF). Folding starts from the unfolded conformation (**left panel**). Next, one of the loops is formed (**middle-left panel**) followed by the formation of the second loop (**middle right panel**). Finally, both of the tails are structured, and the order of the tail attachment differs between simulations. The placement of the tails leads to the native structure (**right panel**).

## 4. Conclusions

Understanding *how* and *why* there are knots in proteins has been a challenging question in biology for the last few decades. One of the necessary steps to solving this question is to efficiently distinguish and characterise different types of entanglement. The knotoid approach makes significant progress towards characterising the entanglement of knotted proteins as the knotoid distribution gives a topological and geometric description of an open-ended curve, richer than any closure method can give [2]. However, until now, there has been no systematic way to extract information from this distribution about specific geometric features of a curve, with the exception of very recent work by a subgroup of the authors [37], in which the topological complexity given by the $f$-distance is used to separate deeply knotted trefoil proteins from shallow ones.

KnotoEMD provides a sophisticated tool able to precisely and efficiently compare the entanglement of open knotted curves. Remarkably, with KnotoEMD we are able to easily detect local geometric information within curves sharing the same global topology, which other methods have struggled to do. Importantly, we show that this geometric information can be used to reveal different folding pathways of trefoil proteins directly from their native states. Our methods are, of course, not restricted to studying folding hypotheses of proteins, as the topological and geometric information can be applied to other problems in protein science. In fact, they can be applied more generally to the study of entangled open polymers, such as DNA or RNA.

**Author Contributions:** A.B. and D.G. conceived the research. A.B., D.G., O.V. and N.Y. designed the research. A.B., D.G., B.I.M., O.V. and N.Y. performed the research. P.D.-T. performed the simulations. O.V. and N.Y. contributed equally to this work. All authors have read and agreed to the published version of the manuscript.

**Funding:** A.B., D.G. and P.D.-T. thank the COST Action European Topology Interdisciplinary Action (EUTOPIA) CA17139 for supporting collaborative meetings of the authors. A.B. acknowledges funding from the Hooke Fellowship. O.V. is supported by the EPSRC studentship EP/N509711/1 and the EPSRC grant EP/R018472/1. N.Y. is supported by The Alan Turing Institute under the EPSRC grant EP/N510129/1.

**Institutional Review Board Statement:** Not applicable.

**Informed Consent Statement:** Not applicable.

**Data Availability Statement:** The data is available on the GitHub repository: https://github.com/nyerolemou/proteins-knotoEMD (accessed on 1 June 2021).

**Acknowledgments:** The authors thank Andrzej Stasiak for helpful conversations.

**Conflicts of Interest:** The authors declare no conflicts of interest.

**Abbreviations**

The following abbreviations are used in this manuscript:

| | |
|---|---|
| PDB | Protein Data Bank |
| 2D | 2-dimensional |
| 3D | 3-dimensional |
| 1-L | Single-loop open trefoil configuration |
| 2-L | Double-loop open trefoil configuration |
| AOTCases | N-acetylornithine transcarbamylase |
| OTCases | Ornithine Carbamoyltransferase |
| $C\alpha$ | alpha-carbon |
| UMAP | Uniform Manifold Approximation and Projection for Dimension Reduction |
| SMOG | Structure-based Models for Biomolecules |
| DNA | Deoxyribonucleic acid |
| RNA | Ribonucleic acid |
| tRNA | Transfer RNA |
| PL | Piecewise-linear |

## Appendix A. Trefoil Proteins

In our work, we consider all the 517 proteins whose backbones form open-ended positive trefoil knots catalogued in the KnotProt DataBase [2] (as of August 2020).

### *Appendix A.1. Restriction to Knot Core*

The knot core of an open knot is defined as the shortest portion of the curve involved in the knot [2,35]. For some of our computations, we only needed to analyse the geometry and topology of knotted cores, rather than of the entire curves. In these cases, we computed the position of the knot core and of the two ends in each backbone with the software Knoto-ID [35], that uses a top-down approach. We then removed both ends from the backbone, leaving a tolerance of one amino acid on each side. The xyz files of each protein backbone and of their cores are available on our GitHub repository [46].

### *Appendix A.2. Clustering by Sequence-Similarity*

A sequence similarity search was performed for each of our 517 proteins using RCSB PDB Search API [49]. Using the similarity scores returned by these searches, we constructed a similarity matrix for the 517 proteins, assuming zero sequence similarity between proteins for which no score was reported. We converted the similarity matrix to a dissimilarity matrix by converting $\alpha\% \rightarrow (100 - \alpha)\%$ and performed single-linkage hierarchical clustering with the function `scipy.cluster.hierarchy.linkage` [50]. We used a distance threshold of 70% dissimilarity to produce a clustering of the 517 proteins. We extracted the nine largest clusters from these clusters to study.

### *Appendix A.3. List of Non-Redundant Proteins*

Among all the 517 trefoil proteins considered, we sometimes used as representatives the list of non-redundant ones given by KnotProt [4]. Note that in this list seven of the proteins are obsolete PDB entries [1]. In each of these cases, we replaced the entry with the corresponding updated PDB version.

The list of the 517 proteins we considered, the list of non-redundant ones and the sequence similarity clustering are all available on our GitHub repository [46].

## Appendix B. Double-Loop and Single-Loop Open Trefoil Configurations

*Appendix B.1. The Twelve 2-L Configurations and Local Moves between Them*

Flapan et al. identify twelve different 2-L configurations forming open-ended trefoils, depending on the choice of several parameters, which regulate the number of twists in the loops and their relative positions. More precisely, the parameters prescribe the following characteristics:

- The mutual positions of the blue end and the two loops (indicated by a string of length two in *R* and *L*).
- The sign (+ or −) of the bottom crossing in each diagram.
- The (signed) number of (positive or negative) twists in the two loops (indicated by two integers *a* and *b*);

The twelve 2-L configurations are shown in Figure A1. We observe that the twelve proposed 2-L configurations can be divided into three groups, where configurations within each group are related by local transformations, as illustrated in Figure A1. Among these three groups, the one whose configurations have the lower average crossing number (those forming the left-hand square in Figure A1) will be called *simple* 2-L configurations.

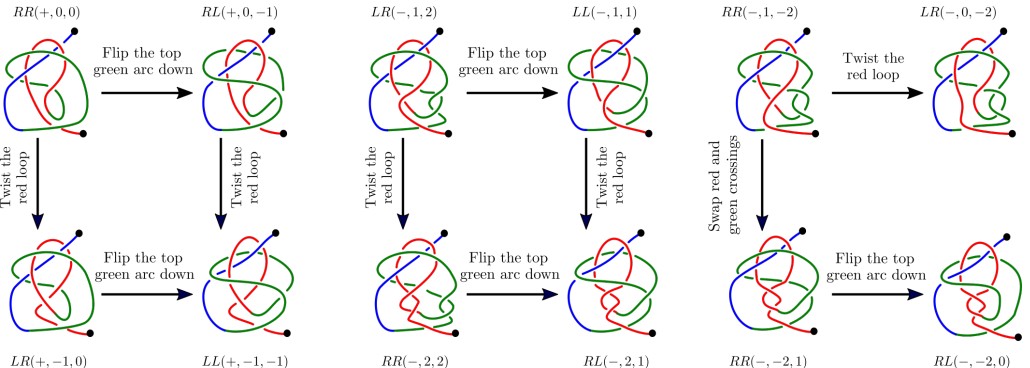

**Figure A1.** 2-L configurations and local moves between them. The 12 2-L configurations differ by the number of twists in the loops and their relative positions. These configurations can be divided into groups related by local transformations that do not change the overall geometry of the curves. The 4 configurations on the left-hand side are the ones with lowest average crossing number. We will refer to them as simple 2-L configurations.

*Appendix B.2. Generating Our Dataset of Trajectories*

Our dataset of trajectories was generated as follows.

- Step 1: create the representative trajectories. For each of the 12 2-L configurations we created two different representative piecewise-linear (PL) curves using the software KnotPlot [51]. In the same way, we created four different 1-L PL curves representing minimal (thus, admitting a projection with only 3 crossings) geometrical embeddings of open trefoils. All the curves were drawn to be quite shallow (i.e., with most of the curve involved in the knot).
- Step 2: take different lengths of each curve. We then subdivide each trajectory in three different ways, to obtain curves of length approximately (here the length is measured as the number of segments in the PL curve) 80, 160 and 240 (this is to match the different lengths of trefoil proteins' knotted cores). In this way, we obtain a total of six different PL curves for each 2-L configuration, and 12 curves representing a 1-L configuration, for a total of 84 curves.
- Step 3: perturb each curve. We then generate 10 different trajectories for each of the 84 curves by performing numerical perturbations. The minimal distance $d_m$ between vertices of each trajectory is determined. Each vertex is perturbed uniformly within a sphere of radius $d_m$ centred at the vertex. This step adds some randomness to a curve

without breaking the geometry of the loops. The perturbation script is available in our GitHub repository [46].

In this way, we end up with 60 different trajectories for each 2-L configuration, and 120 1-L trajectories, for a total of 840 different PL curves forming open ended trefoils with different geometries. The xyz coordinates of all the trajectories in our dataset are available on our GitHub repository [46].

## Appendix C. Computation of Knotoid Distributions and KnotoEMD

### *Appendix C.1. Knotoid Distributions*

From the xyz file of each PL curve (either representing the entire chain of a protein or trajectory, or their reductions to the knot core) the knotoid distribution is computed using the software Knoto-ID [35]. More precisely, a total of 5000 projections are sampled and then subsequently analysed with polynomial invariants to detect their knotoid type. By results in [37], a sample size of 5000 is known to be optimal to well approximate the continuous knotoid distributions.

### *Appendix C.2. KnotoEMD*

KnotoEMD is defined as the earth mover's distance between their knotoid distributions, with cost matrix given by the $f$-distance. Pairwise distances are computed using Python EMD wrapper PyEMD [25,52]. The cost matrix containing the pairwise $f$-distances between knotoids with minimal crossing number at most 6 is obtained from results in [37]. The cost matrix and the KnotoEMD computation script are available on our GitHub repository [46]. A small number of trefoil proteins and some of the most complicated 2-L trajectories have knotoid distribution containing a few occurrences of knotoids with minimal crossing number larger than 6. As the current classification [53] of knotoids is restricted to minimal crossing number at most 6, these entries are collectively catalogued as "'unknown". The $f$-distance between an unknown and any other knotoid is set to 2. As the probability of occurrence of unknowns for the curves considered is very low, the overall aspect of our computations does not change for different choices of the distance between unknowns and other knotoids, such as for example 4 or 6.

### *Appendix C.3. Distance Matrices*

This section contains a description of the various KnotoEMD matrices we computed. Figure A2 shows heat maps for pairwise distances of entire chains and knot cores of trefoil proteins. Similarly, Figure A3 contains pairwise distances of our generated trajectories (restricted to the core), and distances between these and knot cores of trefoil proteins.

In all the relevant plots, proteins with similar sequences are placed closed to each other. The complete ordering and sequence similarity clustering are available on our GitHub repository [46]. The generated trajectories are ordered as follows.

1.  The simple 2-L trajectories, in the following order:
    - the 60 trajectories for RR(+,0,0);
    - the 60 trajectories for RL(+,0,-1);
    - the 60 trajectories for LR(+,-1,0);
    - the 60 trajectories for LL(+,-1,-1).

2.  The first group of complex 2-L trajectories, in the following order:
    - the 60 trajectories for LR(-,1,2);
    - the 60 trajectories for LL(-,1,1);
    - the 60 trajectories for RR(-,2,2);
    - the 60 trajectories for RL(-,2,1).

3.  The second group of complex 2-L trajectories, in the following order:
    - the 60 trajectories for LR(-,0,-2);
    - the 60 trajectories for RR(-,1,-2);

- the 60 trajectories for RR(-,-2,1);
- the 60 trajectories for RL(-,-2,0).

4. The 120 1-L trajectories.

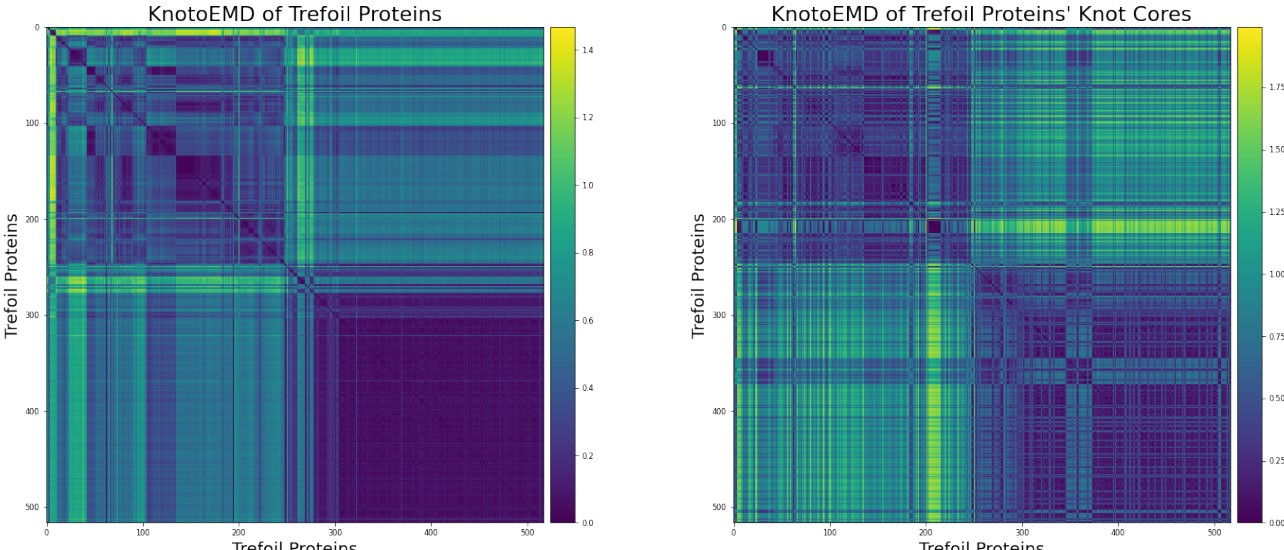

**Figure A2.** Pairwise KnotoEMD on trefoil proteins. On the left, knotoid distributions are computed from the entire chain, while on the right, from the knot core of each protein. Proteins are ordered according to sequence similarity: proteins in the same sequence similarity cluster are placed close to each other. The top noisy square represents pairwise distances between singletons (i.e., proteins that are not similar to any other protein considered). Note that pairwise distances between proteins in the same sequence similarity group are very low, showing that our distance recovers clustering by sequence similarity. This remains true when we only analyse the core, although the results are noisier in this case.

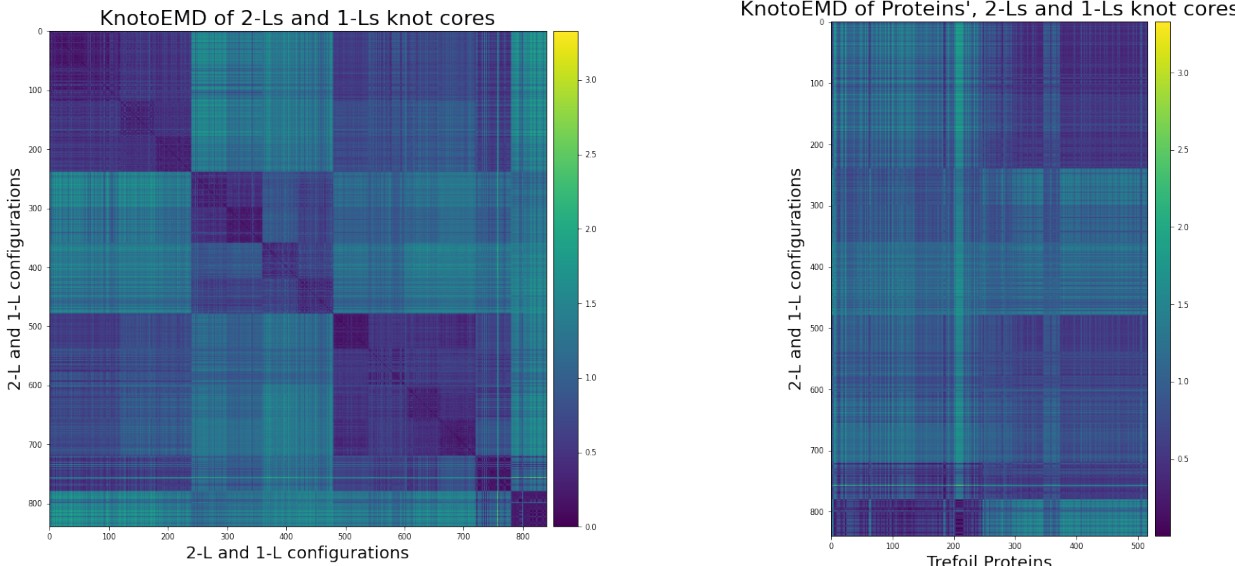

**Figure A3.** KnotoEMD of 2-L configurations, 1-L configurations and trefoil proteins cores. All the pairwise distances shown here are computed from the knotoid distributions of the knot core of each curve. Pairwise distances of our generated trajectories are shown on the left, and distance between trajectories and trefoil proteins are on the right. Note that KnotoEMD clusters each group of 2-L configurations, and the group of 1-L configurations. Similarly, distances from trajectories to proteins respect both the sequence similarity subdivision of proteins and the subdivision of trajectories.

*Appendix C.4. UMAP Projections*

To visualise the spatial structure of the metric space of proteins equipped with KnotoEMD, we used the nonlinear dimension reduction technique UMAP [40] on the KnotoEMD distance matrices. This dimension reduction technique requires choices of four parameters:

- `n_neighbors`: a constraint on the size of local neighbourhood considered in the dimension reduction.
- `min_dist`: the minimum distance separating points in the reduced dimension space.
- `metric`: the metric used to compare the points of the input space (in our case rows of a large distance matrix).
- `n_components`: the target dimension of the low dimensional space to which we project.

The parameter `n_neighbors` controls the emphasis on local vs global structure preservation in dimension reduction, and was set to 60 representing $\sim 10\%$ the number of points. The parameter `min_dist` affects the ability for points to tightly pack in the low dimensional embedding, and was set to 0.3 to balance fine and broad topological structure. Our plots were not highly sensitive to the choice of either the `n_neighbors` or `min_dist` parameter. We set `metric = 'precomputed'` to indicate that our input is a distance matrix rather than a sample-feature matrix. We produced both 2 and 3 dimensional plots [46].

*Appendix C.5. The Knotted/Unknotted Homologous Pair*

Note that KnotoEMD captures information on the topology and geometry of open-ended knots at the same time. Indeed, while it efficiently detects different structural and geometrical features among proteins forming open trefoils, it is at the same time able to distinguish proteins with almost superimposable structures differing only by a strand passage, as in the case of the pair knotted AOTCases/unknotted OTCases [4], shown in Figure A4. The complete distance matrix and the order of the proteins considered are available on our GitHub repository [46].

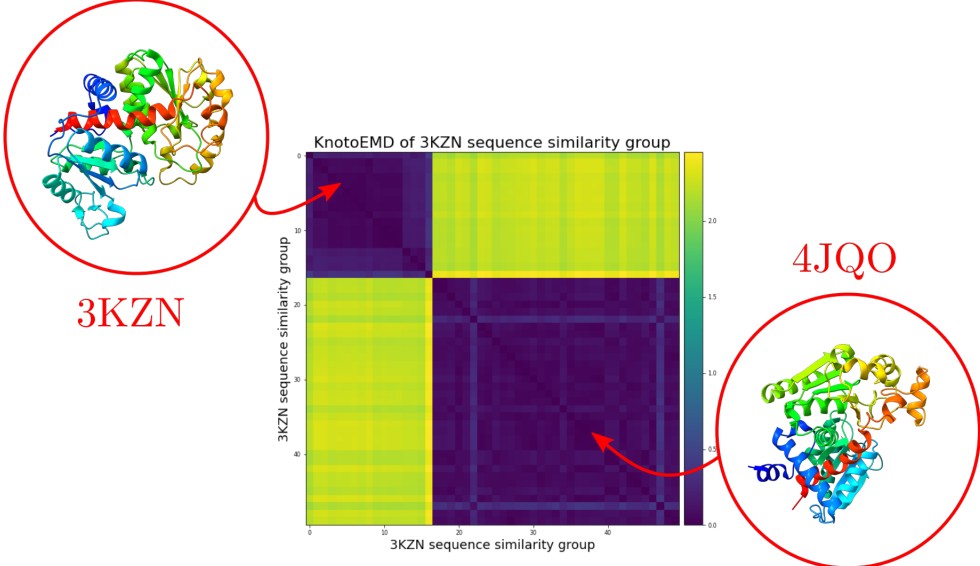

**Figure A4.** KnotoEMD of knotted and unknotted proteins with sequence similar to 3KZN. The deeply knotted AOTCases with PDB entry 3KZN has sequence highly similar to several unknotted proteins. The heat map shows pairwise distances for proteins in this group. The first 17 entries represent the knotted ones, while the following ones are unknotted. The pair knotted AOTCases/unknotted OTCases, whose structures are almost superimposable [4], is highlighted.

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
