# Peer review of "A Topological Selection of Folding Pathways from Native States of Knotted Proteins"

_symmetry, doi:10.3390/sym13091670_

Round 1
Reviewer 1 Report
Manuscript submitted by Barbensi et. al "A topological selection of folding pathways from native states of knotted proteins" in symmetry (symmetry-1347985). It requires revision to improve result description, analyses and interpretation, as well as elimination of grammatical errors. I have been carefully read the manuscript and there are many issues in the manuscript, as explained below:
- Many palaces unnecessary capital letters e.g. Carbonic Anhydrases. Author should avoid unnecessary capital letters.
- A lot of abbreviation used in the current study and advised to the authors should provide the in the list of abbreviation.
- Author should add more section related to the energy change in the knotting.
- Page 9, 302-303: “There is a consensus that multiple folding pathways are possible for shallow proteins”. For that author can take help and cite the paper. “RSC Advances 2015, 5 (26), 20115-20131”.
Author Response
"Please see the attachment."

Reviewer 2 Report
Very interesting presentation of ideas and proposed conclusions. Well
supported by discussion though more detail will be required by some
readers.
Reviewer 3 Report
Understanding the folding and the structures of the class of proteins that are rare but contain a topological knot in their backbone is not novel since many other methods have been developed. Nevertheless, the authors present an additional mathematical approach based on KnotoEMD as a statistical analysis of distance defined on distribution of protein backbone projections.
The introduction section reports the state of the art and advantages of the methodology respect to the state of the art.
Then, the methodology describes the potentiality of KnotoEMD.
The appendix describes step by step all the mathematical method to determine the folding pathways that bring to the knotted proteins.
The results are clearly presented and supported by the figures that show the local geometric features suggesting the different folding pathways for the trefoil proteins.
The quality of the figures needs to be improved. On the other hand, the figures’ caption well describe the figure panel.
The conclusion is in agreement with the results reported.
More in general the paper is well written and easy to read.
I’m glad to consider this manuscript for publication.

Author Response
"Please see the attachment."
